# Classification and Multi-Functional Use of Bacteriocins in Health, Biotechnology, and Food Industry

**DOI:** 10.3390/antibiotics13070666

**Published:** 2024-07-18

**Authors:** Miguel Angel Solis-Balandra, Jose Luis Sanchez-Salas

**Affiliations:** Department of Chemistry and Biological Sciences, Sciences School, Universidad de las Americas Puebla, Ex-Hacienda de Sta., Catarina Martir s/n, San Andres Cholula, Puebla 72810, Mexico

**Keywords:** bacteriocins, *lantibiotics*, *colicins*, *microcins*, antimicrobial resistance, antimicrobial coating, *nisin*

## Abstract

Bacteriocins is the name given to products of the secondary metabolism of many bacterial genera that must display antimicrobial activity. Although there are several bacteriocins described today, it has not been possible to reach a consensus on the method of classification for these biomolecules. In addition, many of them are not yet authorized for therapeutic use against multi-drug-resistant microorganisms due to possible toxic effects. However, recent research has achieved considerable progress in the understanding, classification, and elucidation of their mechanisms of action against microorganisms, which are of medical and biotechnological interest. Therefore, in more current times, protocols are already being conducted for their optimal use, in the hopes of solving multiple health and food conservation problems. This review aims to synthetize the information available nowadays regarding bacteriocins, and their classification, while also providing an insight into the future possibilities of their usage for both the pharmaceutical, food, and biotechnological industry.

## 1. Introduction

The term bacteriocins refers to proteins or peptides of ribosomal production that display either inhibitory or lytic activity against bacterial cells, whether they are of the same genus of the producing bacteria or a closely related genera, and they even cover a wide spectrum of microorganisms in some cases [1]. These products might have post-translation modifications or perform their function with their original structure [2]. The first report of a characterized bacteriocin dates to 1925, being named colicin, a name given based on the microorganism from which it was isolated: *Escherichia coli* [1,2].

These biomolecules are products of the secondary metabolism of several bacterial genera and take part in the elimination of competing microorganisms present in the same ecological niche. However, the production of bacteriocins is an energy- and nutrient-demanding process, so not all strains conduct this process continuously; in fact, the production of these metabolites responds to genetic self-regulation systems known as quorum sensing mechanisms (QS) [3].

As more information regarding bacteriocins has become available, researchers have defined different classification criteria and/or systems for these metabolites; the first attempts at classification date to 1993, when bacteriocins used to be divided into groups based merely on their physicochemical properties, but as bacteriocins have become more understood by the professionals of the area, classification has evolved to consider the chemical structure, producing organism, molecular weight, and other characteristics that have allowed for better segmentation of the kinds discovered today [4].

The fact that these metabolites have antimicrobial activity against either pathogenic microorganisms or microorganisms known as deteriorators gives them high importance for the pharmaceutical and/or biotechnology industry. As of today, bacteriocins are well renowned for their antimicrobial potency at relatively lower concentrations compared to other antimicrobials; for example, bacteriocins produced by lactic acid bacteria like nisin and the colicin E1 can inhibit the growth of pathogenic bacteria such as *Clostridioides difficile* and *E. coli* from food surfaces, and, moreover, bacteriocins have shown efficacy against strains like methicillin-resistant *Staphylococcus aureus* (MRSA), Vancomycin-resistant enterococci (VRE), and *Salmonella enterica* on the clinical field [5]. Such bacteriocins have demonstrated their effectiveness in animal models; however, sufficient information on their stability, toxicity, and potential side effects in humans is lacking [6]. Still, their potential use in the area of medicine exists, and these problems can be solved.

Despite of the relevance of these biomolecules for multiple industries, information available today on the topic, while not scarce, is usually spread over multiple sources regarding specific kinds of bacteriocins or concrete applications; therefore, this review aims to synthetize the information available on bacteriocins, with the objective of increasing the ease of understanding of these biomolecules for students or professionals of the field.

## 2. Bacteriocin Classification

Regarding the classification of bacteriocins, it has been fluctuating and evolving in parallel with the increase in knowledge of these biomolecules. Initial classifications dating back to 1993 and 1995 were based solely on physicochemical properties such as thermostability and molecular weight. Later, some bacteriocins were classified by their sensitivity to enzymes, their post-translation modifications, or the presence of specific functional groups [4]. This classification system, although no longer recognized, is still used as a basis for the current system, which was developed by various investigations that took place from 2012 to 2018. From this research, the first branching criterion corresponds to the producing microorganism, for which we have Gram-positive bacteriocins and Gram-negative bacteriocins, with some authors also considering those produced by archaea organisms like the halocins (classification not adopted by all authors) [4,7].

### 2.1. Gram-Positive Bacteriocins

As the name implies, Gram-positive bacteriocins are those whose production comes from Gram-positive bacterial genera such as *Lactobacillus* and *Staphylococcus*. Gram-positive bacteriocins are subclassified into three groups based on the presence of post-translation modifications (class I) or their absence (class II), while having a third class which presents tertiary structure [2,4]; the general characteristics of each class are detailed below and in Figure 1.

Class I: Also known as lantibiotics, they are molecules with a molecular weight of less than 5 kDa, thermostable and with a high abundance of post-translation modifications. Within their structure, there is a high proportion of certain amino acid structures such as lanthionine and methyl-lanthionine (giving its name to this class); as well as unsaturated amino acids, this amino acid composition gives them the capacity of forming intramolecular ring structures, generally via di-sulfide bonds [7]. This class has two additional subdivisions:Class Ia: grouping of structures that are polar, with a positive net charge [2].Class Ib: which includes those that lack a net charge or have a negative net charge [2].

Class II: The second class of bacteriocins of Gram-positive bacteria covers equally small molecules, this being <10 kDa. This class is subdivided into four subclasses. All of them share the characteristic of having minimal, or even no, post-translation modifications [4,8].
Subclass IIa: covers peptides that have activity against *Listeria* (pathogenic bacterial genus that causes food-borne disease) [2].Subclass IIb: corresponds to peptides that act in a dimeric conformation, in which two unaltered peptides act synergistically to achieve the antimicrobial effect [2,4].Subclass IIc: includes peptides with a circular structure [4].Subclass IId: includes linear peptides that do not have activity against *Listeria* [4].

Class III: Encompasses proteinaceous bacteriocins with a high molecular weight (>30 kDa), which have the characteristic of possessing a complex structure linked to their function, a fact that makes them thermolabile. This group has the particularity of encompassing some bacteriocins that are also produced by Gram-negative bacteria under some circumstances, such as klebcin [2].

### 2.2. Gram-Negative Bacteriocins

Regarding the bacteriocins produced by Gram-negative microorganisms, the classification is limited to two groups given the little information available to date on this class of biomolecules. Gram-negative bacteriocins are, in general, mostly isolated from producing strains of *E. coli* or from other enterobacteria. The two types of bacteriocins that make up this group are Colicins (the class to which the colicin bacteriocin described above belongs) and microcins; their differentiation is based mainly on their molecular weight [2,4]. However, there is a third type, which is not yet fully characterized, which will be addressed as a pseudo-third type (Figure 2).
Colicins: These are biomolecules with a molecular weight of 30–80 kDa, generally produced by strains of *E. coli* that harbor a plasmid called colicinogenic, and some authors propose the subdivision of this group into two classes: Colicins produced by *E. coli* specifically, which is additionally subdivided according to the type of plasmid from which they originate, and another group that includes the colicins produced by other member of *Enterobacteriaceae*; however, this classification is not yet adopted by all authors [2,9].Microcins: These include low-molecular-weight bacteriocins, which are peptides of 1 to 10 kDa with a highly stable molecular structure that are active at a wide pH range, not very sensitive to the activity of proteases (a highly desirable characteristic in microbiomes such as the human digestive system), and resistant to temperature changes [9].

These bacteriocins are encoded in the bacterial genomic DNA, unlike colicins [2,9]. Like the previous type, this classification also has a subclassification that has not been fully adopted yet and is based on molecular weight: Class I (<5 kDa) and Class II (5 to 10 kDa) [2].
Phage Tail-Like Bacteriocins: The third hypothetical type of Gram-negative bacteriocin, these are molecules that hypothetically have antimicrobial activity based on their structure, but there is still not much information about them [9].

### 2.3. Alternative Classification Criteria

Aside from the classification based on the cell wall characteristics of the bacteriocin-producing strain, authors have proposed different criteria for the classification of these biomolecules, ranging from their structure to their amino acid sequence and physicochemical properties [10].

The structural classification has two main kinds: The cyclic-structure bacteriocins like the “lasso peptides” (known for their tertiary structure of intramolecular bonding), and the linear-structure bacteriocins like lactococcin, a linear lantibiotic. Meanwhile, classification via animo acid sequence is like the structural one, being based on the secondary structure that certain amino acid sequences form, classifying the bacteriocins via the presence of α-helixes, β-sheets, loops, and other complex structures (like cycles). On the other hand, physicochemical classification works by setting boundaries on the values of properties such as the molecular weight to group bacteriocins [7].

### 2.4. Classification of Bacteriocins via RiPP Nomenclature

Another alternative way to classify bacteriocins relies on their denomination as members of the “ribosomally synthesized and post-translationally modified peptides” group. This fact allows bacteriocins to be classified following the recommendations made by Paul G. Arnison and collaborators in 2012 regarding the nomenclature and classification of these natural products [11].

RiPPs are classified into groups following both structural and synthesis (post-translational modification enzymes) characteristics. However, it is important to remark that while all bacteriocins are RiPPs, not all RiPPs are bacteriocins, and due to this not all groups of RiPPs contain bacteriocins; therefore, only those that contain them will be listed below [11].
Lanthipeptides: These are the RiPPs that contain lanthionine; lantibiotics like nisin fall under this category. Lanthipeptides are further subclassified into four different classes according to the specific enzymes used on their synthesis (Lan synthetases) [11].Linaridins: These RiPPs share characteristics with the lanthipeptides described above; they differ in their synthetic pathway, and the main member of this class is cypemycin [11].Proteusins: This group encompasses peptides of complex structure and with a high abundance of both nonproteinogenic and D-configurated amino acids; an example of a type of peptide belonging this class is the polytheonamides [11].Linear azol(in)e-containing peptides (LAPs): LAPs are RiPPs characterized by the presence of thiazole and (methyl)oxazole heterocycles; one example of this kind of RiPPs is microcin B17 [11].Cyanobactins: This class of RiPPs groups exclusively short peptides synthetized by cyanobacteria that exhibit diverse structures but similar gene clusters [11].Thiopeptides: Their main characteristic is the presence of a thiopeptide macrocycle alongside the abundance of dehydrated amino acids; this class of RiPPs is commonly synthetized by *Bacillus* and *Staphylococcus* bacterial species, with the most representative kind of bacteriocin being micrococcins [11].Bottromycins: These peptides are named after the microorganisms from which they were originally obtained (*Streptomyces bottropensis*). Their main characteristic is the display of a macrocyclic amidine and a decarboxylated C-terminal thiazole on their structure. One of the most renowned members of this class is the bacteriocin bottromycin A2, which displays a strong antimicrobial effect against pathogenic bacteria [11].Microcins and Colicins: Similarly to the classical bacteriocin classification, microcins and colicins are the bacteriocins produced by enterobacteria, mainly *E. coli*, that are differentiated and classified with the same criteria as shown in Section 2.2 of this review [9,11].Lasso peptides: These are generally characterized by the presence of a structure named the “lasso fold”, which is formed by the interaction of a terminal macrolactam ring with the C-terminal tail of a short peptide; lasso peptides tend to be stable bacteriocins generally produced by actinobacteria and proteobacteria, and one example of this kind of RiPP is microcin J25 [11].Sactibiotics: Sactibiotics are a subclass of the group of RiPPs named sactipeptides, characterized by the linking of cysteine residues to the alfa carbons of other amino acids. Sactibiotics are generally produced by bacteria from the *Bacillus* genus [11].Bacterial head-to-tail cyclized peptides: The last class of bacteriocin RiPPs encompasses the peptides that form a peptide bond between the C-terminal amino acid and the N-terminal amino acid, and they differ from other cyclic RiPPs due to their size (generally larger) and their synthesis machinery; the main representative of this class is enterocin AS-48, which shows both antimicrobial activity and other desirable biological effects [11].

However, it is important to highlight that the classification of bacteriocins is still a fluctuating topic, with authors recommending diverse ways of classifying these biomolecules that often overlap. While the accuracy and importance of the different systems cannot be undermined, one of the more pressing issues on the study of bacteriocins to be solved in the near future is the lack of a universally adopted system, making it one of the main areas of opportunity for researchers to either generate a novel classification based on existing criteria or to adapt and use those available today [12].

## 3. Mechanism of Action of Bacteriocins

In this context, it is important to highlight that there is no certainty about the mechanism of action of all the bacteriocins described today. An example can be observed in bacteriocins such as PLNC8, which has a confirmed inhibitory capacity against *Helicobacter pylori*, but its mechanism of action is unknown [13]. On the other side, there are groups of bacteriocins whose mechanisms of action are fully described. An example of these groups corresponds to the bacteriocins produced by lactic acid bacteria, such as *Lactobacillus*, also known as LAB-bacteriocins [4].

### 3.1. LAB-Bacteriocins

This class of bacteriocins belongs to the Gram-positive bacteriocins; among this group, the most common and the best-known type is the lantibiotics (Class I) [4,14].

Lantibiotics have demonstrated two different mechanisms to exert their bacterial lysis function: The first is the disruption of cell wall synthesis; the second is the formation of pores.
Disruption of Cell Wall Synthesis: In this mechanism, various lantibiotics show antibiotic activity through two main ways of inhibiting the synthesis of the cell wall: The first is binding to lipid II (an important intermediate in the trans-glycosylation reaction); an example of a bacteriocin that uses this mechanism is gallidermin [4,15]. The second mechanism is the blocking of the incorporation of glucose and D-alanine in the precursors of cell wall molecules, thus inhibiting the synthesis of peptidoglycan; however, it has been demonstrated in studies by various authors that this mechanism is also dependent on the availability of lipid II (Figure 3) [4].Pore Formation: The second way in which lantibiotics conduct their bactericidal activity relies on their ability to attack the integrity of the cell membrane (Figure 4).

Within this mechanism, there are two models currently proposed: The first is the “barrel-stave” model, in which the bacteriocin binds in parallel to the bacterial membrane; this union causes, through its difference in charges, a loss of membrane potential and an accumulation of water, alongside the formation of pores, and all this leads to the leakage of solutes and biomolecules from the cytoplasm to the external medium [4,15].

The second model is the “wedge” model, in which the interaction of the bacteriocin occurs in a trans-membrane manner, via the interaction of the charged components of the bacteriocin with the polar head of the lipid bilayer and the interaction of the peptide chain with the non-polar tail of the lipid acid. This insertion of the bacteriocin generates deformations in the membrane and fissures [4]. It has been noted that pore formation can be mediated by binding to lipid II as well [4,8].

### 3.2. Colicins

On the other hand, a group of bacteriocins that has also been adequately characterized is the colicins, biomolecules produced by *E. coli* and other enterobacteria which are specialized in the elimination of other Gram-negative bacteria [12].

The mechanism of function of these bacteriocins is based on their structure. These colicins generally present three “domains” that each have a function: The first is an antigen-like recognition section for anchoring to the cell membrane, a mechanism like an antibody; the second domain is responsible for the introduction of the bacteriocin to the target bacterial cell. Finally, the last domain performs a toxic function. Currently, there are three mechanisms described for the “toxic” domain [12]:Formation of voltage-dependent pores in the inner membrane.Nuclease activity against bacterial genetic material.Inhibition of peptidoglycan synthesis [12].

However, it is important to clarify that the accuracy of how *colicins* exert these mechanisms may vary (Figure 5) [12].

## 4. Bacteriocin-Producing Bacteria Genera

As already mentioned, the production of bacteriocins is a natural process of various bacterial genera in response to competitors in the microenvironment. This opens the door to the assumption that this metabolic activity is common in most bacterial genera discovered today, but this has not been confirmed [1]. However, research has managed to find bacterial species and/or groups that are certain to produce at least one bacteriocin of any type, thus being a fundamental factor in the conformation of the micro-environments where these microorganisms grow [1].

The first group corresponds to the Enterobacteriaceae family, where we can find species such as *E. coli*, *Enterobacter* spp., and *Klebsiella* spp., among others. These groups are recognized to produce bacteriocins of the colicin type, especially *E. coli*, or microcins in the case of the rest of the enterobacteria [12].

On the other hand, a group of bacteria well known for their production of bacteriocins corresponds to lactic acid bacteria, that, in addition to the production of non-protein antimicrobial substances such as lactic acid, are recognized producers of lantibiotics [16]. Within this group, we find bacteria of the genus *Lactobacillus*; species of this genus are used nowadays as oral probiotics, which are commonly recommended to patients after antibiotic therapies or in cases of stomach infections due to resistant pathogenic microorganisms, with the aim of recovering the balance of the microbiome by taking advantage of their ability to secrete bacteriocins that attack colonizing foreign microorganisms [17,18].

Additionally, recent research has discovered the production of bacteriocins by the *Bacillus*, *Staphylococcus* and *Streptococcus* (in particular, beta-hemolytic species) genera, whose bacteriocins have recently been isolated and are in the process of being developed for a possible biotechnological application [15,19,20].

Finally, it should be noted that the procedure for detecting bacteriocins in bacterial isolates can be cumbersome and repetitive, which is why in recent years “machine learning” mechanisms have been developed to assist in the detection of genes that codify for the synthesis of these bioproducts [21].

Regarding the genes involved in the production of bacteriocins, two types have been reported, the first being chromosomal gene clusters known as “operons”; an example of them, is the “thermophilin 13 operon” that allows certain strains of *S. thermophilus* to produce the bacteriocin thermophilin [22]. The second type of bacteriocin-encoding genes are related to the presence of “orphan genes”, which are single genes that allow by themselves the production of a certain type of bacteriocin; an example of this can be found in certain strains of *L. plantarum*, which can carry orphan genes like PlnJ and PlnNC8, and have been noted to be closely related to other bacteriocin orphan genes from related strains, suggesting that the orphan genes probably come from a common ancestor and are transmitted via plasmids or another gene transferring strategy [23].

### Bacteria Source and Selection

Having discussed the bacteria genera that could produce bacteriocins, the next question to address is the source from which said bacteria could be isolated from. Multiple studies have been successful in isolating potential bacteriocin-producing strains from natural sources as river water, grass silage, and soil [24]; additionally producing strains can also be found on prepared food items, like the Korean traditional Kimchi, dairy items as cheese, milk, and buttermilk, and a large variety of other food items [25].

Another source from which researchers have been able to recover bacteriocin-producing bacteria is samples of healthy microbiomes, like those taken from either the gut or the oral cavity of healthy individuals, from which multiple bacteria species known for producing bacteriocins are found, generally enterobacteria like *E. coli* or *Enterobacter* spp. [26].

## 5. Isolation of Bacteriocins for Their Use

Regarding the isolation and use of bacteriocins, the methodologies for detection, the determination of action spectrum, isolation, and subsequent characterization have been evolving in parallel with the knowledge about these molecules.

Upon initial assessment, the detection of strains that produce a bacteriocin of biotechnological interest is carried out via various methods, among which are the following: the point inoculation method, cross-streak method, radial-streak method, agar insert method, disk diffusion method, Oxford cup method, and diffusion-well method [27].

These methods are based on the inhibiting of the growth of “indicator” strains caused by the presence of a strain with possible bacteriocin production or using liquid culture supernatants after centrifugation (known as Cell-Free Supernatant or CFS), which can be placed in contact with the indicator strains using various vehicles [27].

Aside from the conventional methods described before, more recent research has been able to develop methods of detecting possible bacteriocin-producing strains using molecular methods that allow the detection of genes or gene clusters that code to produce these biomolecules [21].

Once the identification of a bacterial strain that produces a bacteriocin of interest is achieved, extraction and purification can proceed. The first step is the cultivation of the producing strain in an appropriate liquid medium, from which the CFS will be obtained, where several metabolic products produced by the bacteria, like the bacteriocins, are found [18].

Subsequently, this CFS is subjected to various purification methods to recover the bacteriocin in question in the purest form possible. Among the methods used today, the most common are Ion exchange chromatography, gel chromatography, HPLC, reverse-phase chromatography, and solvent fractionation, among others that allow the separation of the component of interest from contaminants and/or impurities, as well as other elements of the culture medium. All these methods have been described as effective by diverse research, as reported by Ye and collaborators [18].

Finally, after obtaining a purified bioproduct, the characterization methodologies can be applied. These tests are conducted with the purpose of understanding the molecular structure of the bacteriocin (Mass Spectroscopy or IR) and knowing its stability (Enzymatic sensitivity tests, stability in pH gradient, and thermostability, among other tests). These tests are conducted to outline the conditions under which the product could be used to perform its activity against microorganisms of medical interest, act as food preservatives, and/or be used in the biotechnological sector (Figure 6) [18,28].

## 6. Potential Uses of Bacteriocins

Regarding the applications that these biomolecules can have, it is extremely important to emphasize their biotechnological and/or healthcare potential when it comes to battling microorganisms with relatively low toxicity compared to regular antibiotics; some of the potential uses are highlighted below.

### 6.1. Fighting Antimicrobial Resistance

The first use that can be given to bacteriocins and the one that quickly comes to mind is medicinal use as antibiotic therapies against microorganisms that are not susceptible to current antibiotics [16].

Within this area, it is important to mention the global problem of antimicrobial resistance, where microorganisms become resistant to drugs to which they were previously susceptible, due to the selection of clones that have mechanisms and/or mutations, which allows them to survive their effects. The conditions in which this phenomenon occurs are normal, but its appearance is accelerated by the indiscriminate and empirical use of antibiotics for treatments, the misuse of them by patients, their use in other activities such as livestock farming, and their non-controlled disposal in natural ecosystems [30].

In this regard, the WHO (World Health Organization) determined a “priority” group which has demonstrated an accelerated development of resistance mechanisms, known as the ESKAPE group, composed of *Enterococcus faecium*, *S. aureus*, *K. pneumoniae*, *Acinetobacter baumannii*, *Pseudomonas aeruginosa*, *and Enterobacter* spp. It is in this area where research on the use of bacteriocins as medicinal therapy becomes important, since they have demonstrated effectiveness in the elimination and/or inhibition of the growth of resistant strains of these microorganisms, which opens the door to their use as part of the efforts to the fight against this global problem [30]. However, a prominent issue yet to be solved with the use of bacteriocins is their safety in human beings.

### 6.2. Preservative Agents

With respect to the food biotechnology sector, recent studies have shown that the presence of non-pathogenic groups of bacteria, such as *Lactobacillus*, on food items plays an important role when it comes to the extension of shelf life [31]. Within this, it has been shown that the presence of strains that produce a type of bacteriocin can inhibit the growth of microorganisms harmful to health on the surface of foods such as cheese, beef, ham, and prepared food items as cheonggukjang (a traditional Korean dish); additionally the usage of nanometric systems that include a bacteriocin in the food item preparation could also provide extremely good effects on prepared drinks such as wines and fruit juices. All the mentioned bacteriocins have proven to extend the item shelf life by around 30 days compared to non-bacteriocin containing items [31,32].

An example of the practical uses of bacteriocins that have been developed over the years is the application of coatings supplemented with *Lactobacillus* strains for food preservation, which have demonstrated effectiveness in inhibiting the growth of *L. monocytogenes*, a pathogenic bacteria known for causing severe food-borne illnesses [31].

### 6.3. Restoration of the Balance of the Microbiota

Another use of bacteriocins to highlight that has been elucidated for these biomolecules is based on their regulatory capacity for the microbiome. The commensal microbiomes of the various areas of the human body, such as the digestive tract, play an important and first-line role in the defense against pathogenic microorganisms [1,33].

However, when the balance between the microorganisms present is lost, either due to prolonged antibiotic treatments, poor diet, and/or the colonization of harmful microorganisms, a condition known as “dysbiosis” occurs, which has been associated with dangerous diseases such as chronic infections by microorganisms such as *C. difficile*, producing a chronic inflammatory disorder and even the development of cancer [33].

Considering this, the development of probiotic formulations based on lactic acid bacteria that produce bacteriocins, as well as the transplantation of a healthy microbiome (commonly through fecal transplants), have gained relevance in the safe and effective treatment of dysbiosis, achieving equal results or even results superior to conventional antibiotic therapy [33].

### 6.4. Other Pharmaceutical Applications

Although bacteriocins are mainly renowned for their applications based on their bactericidal effect, these biomolecules have been shown in recent research to be capable of other desirable effects for use in the pharmaceutical field. Firstly, one of the major fields where bacteriocins could be applied is in anticancer therapies, and research on the field has shown that positively charged bacteriocins could present a selective cytotoxic effect against some cancer cells due to the overabundance of negatively charged compounds in their cell wall [7,10]; while a mechanism has been described, this being the loss of membrane potential and loss of selective permeability, studies on the matter are still in the early stages, and no clinical data about their effectiveness for cancer treatment have been reported [7].

Another possible application lies on the use of certain bacteriocins, mainly enterocins like those produced by *E. faecium* (enterocin CRL35), as antiviral agents, showing promising antiviral effect against clinically relevant viruses like Herpes simplex virus HSV-1 and HSV-2. Meanwhile, another enterocin (AS-48) has even shown some antileishmanial effects while retaining a low toxicity against macrophages in in vitro studies, opening the door to the use of bacteriocins against other kinds of microorganisms than bacteria in the future [10].

It should be noted that the described uses do not cover the absolute map of what these biomolecules can do for biotechnological purposes.

## 7. Nisin: The First Bacteriocin Approved for Use

Speaking in historical terms, the discovery of the first bacteriocin currently approved by the FDA dates to 1928, the same time that Alexander Fleming discovered penicillin; in this year, the scientists Rogers and Whittier reported the ability of a bacterial strain, at that time known as group N *Streptococcus*, to produce metabolites that inhibit the development of pathogens. This biomolecule ended up being named “Group N *Streptococcus* Inhibitory Substance”, which is abbreviated to nisin by adding the suffix -in [34].

Although at that time the activity of bacteriocins against relevant pathogenic microorganisms such as *Mycobacterium tuberculosis* was demonstrated, nisin was determined to be of little use due to its poor solubility and fragility against enzymes [34]. However, in the 1950s, it was deemed a useful food preservative, due to its ability to be added to food items, inhibiting bacterial genera such as *Clostridium*, *Staphylococcus*, *Bacillus*, and *Listeria*, among other Gram-positive bacteria, without altering the flavor of the food, and without entailing apparent adverse effects, a fact that earned authorization by the FDA as a food preservative, with this being the first bacteriocin authorized for use by this institution [25,34].

This bacteriocin, being a desirable product for both the food and biotechnology industries, has been studied over the years, achieving the production of modified nisins (Figure 7 showing the structure of nisin) that are named with a letter code; the modifications are aimed at giving it better physicochemical properties, as well as allowing conjugation to nanometric systems which have extended the spectrum of the action of nisin to Gram-negative microorganisms [25,34].

## 8. New Technological Trends for the Use of Bacteriocins

Another topic to be addressed in this review is the new technological trends that have been developed in the current decade for the use of bacteriocins on a biotechnological level.

The first system used is based on the regulatory aspect of bacteriocin production by microorganisms, that is, the “quorum sensing” system (Figure 8). Numerous studies have sought ways to generate “optimal” conditions that induce the producing bacteria to synthesize bacteriocins for their subsequent recovery [35].

Research has shown that the main factors that cause a bacteriocin-producing strain to synthesize and release a certain bacteriocin into the environment are as follows: the presence of competing strains, a shortage of nutrients, and the presence of sufficient clones of the producing microorganism. Therefore, a bioreactor capable of controlling these factors is an attractive objective for study and development [35].

Finally, another technological trend to optimize the effect and/or expand the spectrum of the activity of bacteriocins that has been used is the association with nanometric conjugate systems [37]. This technological addition is made seeking to emulate natural mechanisms that can be observed in some *L. acidophilus* strains, involving a microorganism that can generate membrane vesicles and use them as a “vehicle” for delivering the bacteriocins it produces [38].

## 9. Production Earnings of Bacteriocins

On the topic of the possible profits that arise from the usage and/or production of bacteriocins, is important to understand that, as of 2019, the antibiotic industry has generated a profit of an estimated USD 59,000 million worldwide, being forecasted to amass a profit of USD 20 million by 2027 [39]. Bacteriocins could pose a cheaper-to-produce alternative to be applied in the same field as the antimicrobials used nowadays, both in healthcare and in the food industry.

In regard to the application of antimicrobials in the food industry, including the usage of these molecules for both the enhancement of production (seen in the applications of antimicrobials either as additives to fertilizers, or as products given to cattle animals) and the preservation of food items (applied or present on different food items, or used as antimicrobials for cooking items) [40,41], bacteriocins or formulation of probiotics could pose a more sustainable option for farmers and sellers alike. On the last point, it is estimated by the Food and Agriculture Organization of the United Nations that around one-third of food items go to waste, meaning that the successful application of bacteriocins as novel and more effective food preservatives would result in a huge economic impact regarding the avoidance of loss of food [42].

With all that has been discussed, the potential application of bacteriocins as antimicrobials could generate an equal, if not larger, income compared to the current antibiotic industry profit, this considering the possibility of cheaper production and the relative safety associated with their use; however, it is important to understand that a drawback to their use nowadays lies in the feasibility of production on industrial levels, as well as their relatively low stability compared to generally used antimicrobials, meaning that research to solve this drawbacks is needed before deploying bacteriocins on the antimicrobial market [43].

## 10. Perspectives

### 10.1. Synergy Studies among Bacteriocins and Classical Antibiotics or Other Bioactive Compounds

Considering everything described on this review, bacteriocins have shown promising characteristics, which opens the possibility of their individual use as antimicrobial agents, but recent research has found that the association of newly described bacteriocins with other bioactive compounds available today shows possible positive synergic effects compared to their basal effects, making said combinations a desirable alternative to either reduce the quantity of bacteriocin used to achieve a desirable effect or to enhance the activity of already-defined antibiotic/antimicrobial consortia; an example of the former can be seen in the research conducted by Soltani and collaborators, which deduced that the use of the bacteriocin reuterin combined with other bioactive compounds such as organic acids showed a synergic effect that allowed for the desired antimicrobial effect on pathogens using a lower concentration of reuterin [5].

On the subject of the enhancement of the activity of bioactive compounds or mixtures already used today, research has shown that the adding of bacteriocins with known antimicrobial effects, such as antibiotics, can amount to a synergy which allows for the treatment of microorganisms which have previously developed some kind of resistance against the known antimicrobial; an example of this could be the use of bacteriocins produced by *E. faecium* alongside antibiotics such as vancomycin and ciprofloxacin against *L. monocytogenes*, which showed an increased effect compared to that of the individual compounds [44]. Moreover, research involving common *Enterococcus* species associated with urinary tract infections has shown that the usage of bacteriocins such as AS-48, which provides effect at concentrations below 10 mg/L, alongside 20 common antibiotics used for the treatment of these infections such as gentamicin and amoxicillin/clavulanate shows a synergic effect that amounts to a 100-fold increase in the antimicrobial minimal inhibitory concentration, a result which is highly promising for the clinical field as it can lead to therapeutic success using fewer antibiotics for these kinds of infections [45].

Another potential approach is the chemical modification of each bacteriocin, which could enhance their activity, and the use of machine learning or artificial intelligence to improve the action on the bacterial target, widen this activity to current resistant microorganisms, and reduce at the same time the toxicity of such compounds.

### 10.2. Synthetic Post-Translational Modifications for Bacteriocins

Following on to the future perspectives on the use of bacteriocins in the therapeutic and/or food preservation field, one of the most interesting, yet somewhat unexplored, possibilities lie in the artificial post-translation modifications that can be carried out on already discovered bacteriocins with the objective of improving their properties and efficacy.

In this regard, the detection of specific enzyme clusters related to certain post-translational modifications like the methylases/methyltransferases (linked to specific methylations that increase cell permeability), prenyltransferases/N-acetyltransferase superfamily (linked to lipidations that increase serum life and improve physicochemical properties), and the rSAM–SPASM enzyme family named spliceases (linked to the introduction of alfa-keto-beta amino acids, which work as protease inhibitors) has opened up the possibility for their use in the modification of already-known bacteriocins [46].

In fact, these post-translational modifications are already being used today to produce “semi-synthetic” bacteriocins such as the class II lantibiotic named actagardine, which was developed as an anticlostridial agent [11]. The possibility of the usage of these enzymes either in recombinant vectors or in in vitro studies could amount to the production of novel bacteriocins or for the enhancement of the action spectrum or physical properties of bacteriocins available today. An overview of the possibility of usage of these enzymes can be seen in Figure 9.

### 10.3. Potential Drawbacks of the Bacteriocins Known Today

While bacteriocins are generally considered safe thanks to studies like those reported by Benitez-Chao and collaborators [47], where bacteriocins like nisin, plantaricin, and the enterocin AS-48 (some bacteriocins well described and used today) have shown no toxic side effects on mouse models at concentrations well over the MIC of clinically relevant microorganisms, they are still not approved in any country for their use on humans to treat infections; this contrasts with the “generally recognized as safe” approbation given by the FDA to bacteriocins produced by lactic acid bacteria for their use as food additives, which has led to their use as a food conservative in 60 countries nowadays [10].

This lack of authorization, and the near impossibility of the administration of bacteriocins orally, due to their sensitivity to proteases and pH changes (in certain cases) present in the GI track, has made their use in antibiotic therapy more difficult to fulfil [10].

## 11. Conclusions

Bacteriocins, although they are not yet a 100% understood topic, are a group of bacterial metabolites of great interest to the pharmaceutical, food, and biotechnology industries. This is due to their ability to disrupt the microbial development of many microorganisms of interest, alongside other pharmaceutical/biotechnological applications that, in addition to their apparent few adverse effects, make them promising biomolecules for both healthcare and food preservation.

It is for this reason that more efforts must be made to be able to take advantage of them optimally, following objectives described in this manuscript like the final outline of a classification system for these biomolecules, the determination of the mechanism of action of the bacteriocins that lack a described mechanism nowadays, the development of new technological approaches to their use, and finally the creation of appropriate safety tests that allow for the authorization for their therapeutic use. However, it is important that their future use is conducted responsibly to avoid, as is the case with antibiotics, them ending up becoming obsolete for the treatment and elimination of the microorganisms against which they have the promise of action.

## Figures and Tables

**Figure 1 antibiotics-13-00666-f001:**
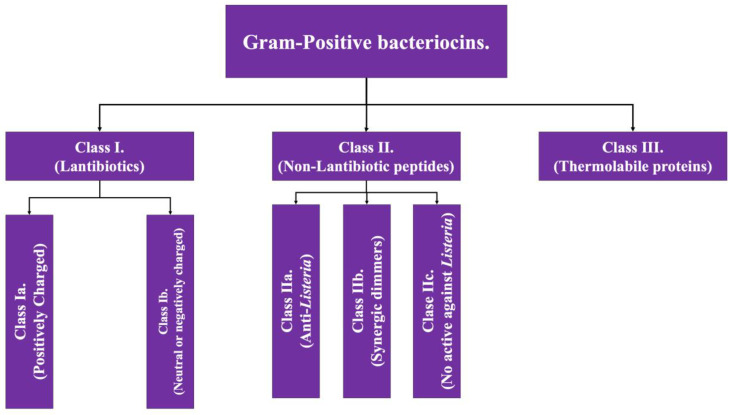
Scheme of the classification of bacteriocins of Gram-positive bacteria. Adapted from Ref. [4].

**Figure 2 antibiotics-13-00666-f002:**
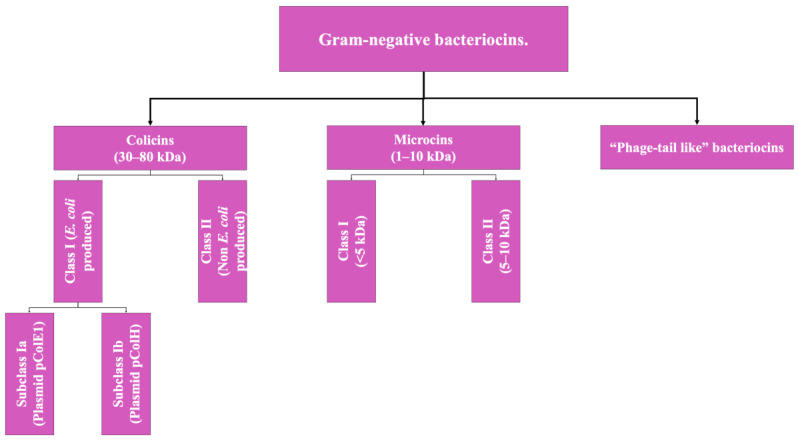
Scheme of the classification of bacteriocins of Gram-negative bacteria. Adapted from Ref. [9].

**Figure 3 antibiotics-13-00666-f003:**
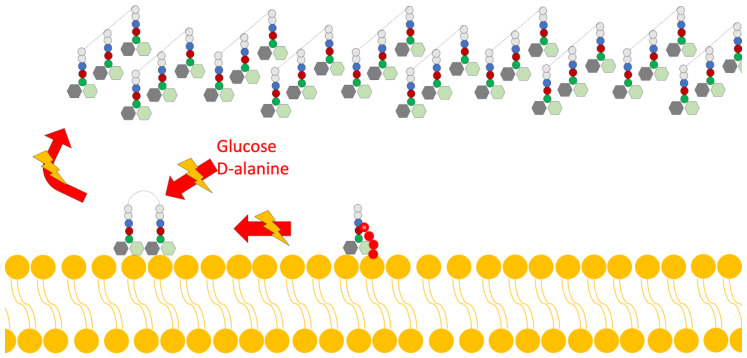
Scheme of the mechanism of action of lantibiotics via the inhibition of cell wall synthesis. Adapted from Ref. [4].

**Figure 4 antibiotics-13-00666-f004:**
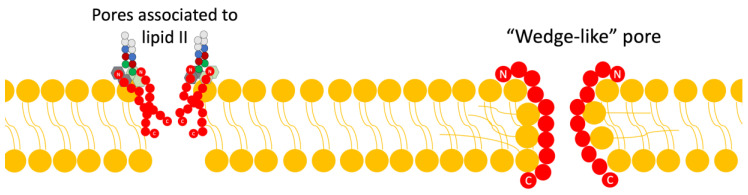
Scheme of the mechanism of action of lantibiotics via pore formation. Adapted from Ref. [4].

**Figure 5 antibiotics-13-00666-f005:**
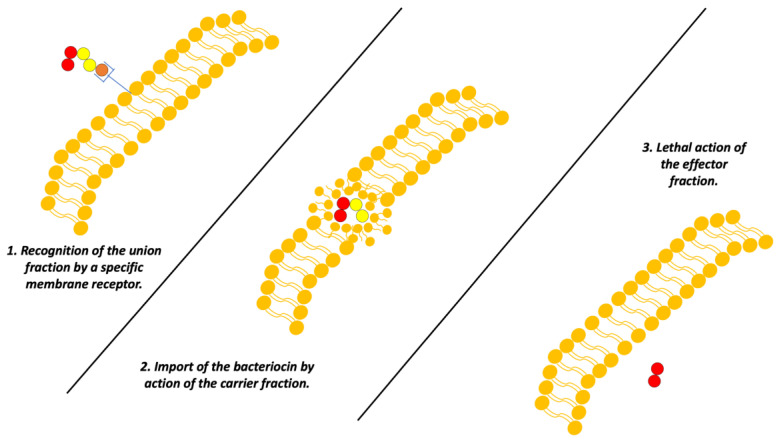
Scheme of the mechanism of action of colicins. Adapted from Ref. [4].

**Figure 6 antibiotics-13-00666-f006:**
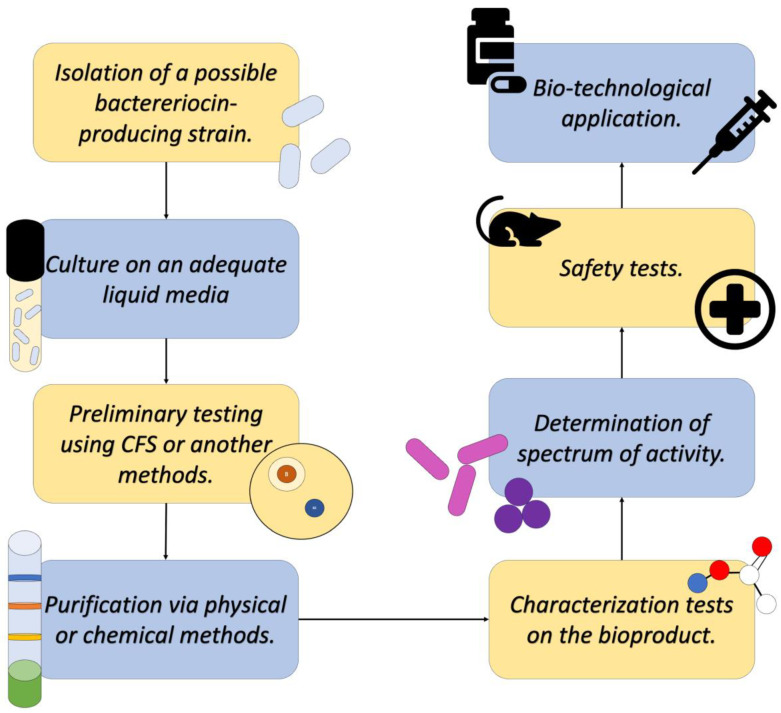
Diagram describing a general process of detecting, isolating, and characterizing a novel bacteriocin. Adapted from Ref. [29].

**Figure 7 antibiotics-13-00666-f007:**
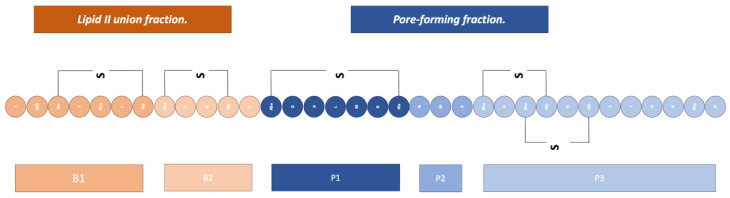
Structure of nisin. Adapted from Ref. [34].

**Figure 8 antibiotics-13-00666-f008:**
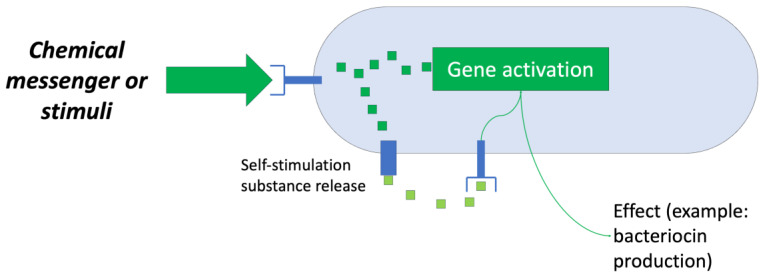
Scheme of a “quorum sensing” system. Adapted from Ref. [36].

**Figure 9 antibiotics-13-00666-f009:**
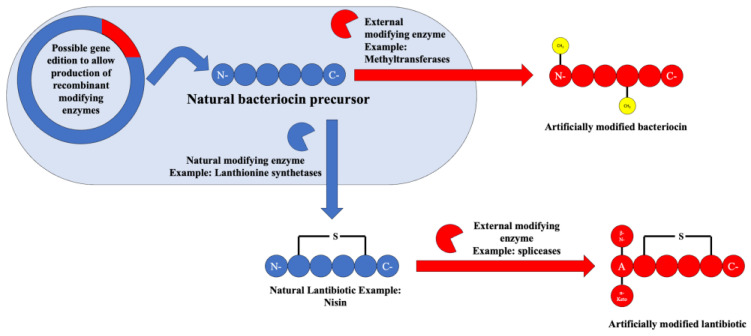
Scheme depicting the possibilities of artificial bacteriocin post-translational modification.

## Data Availability

Not applicable.

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
