# Peer review of "Classification and Multi-Functional Use of Bacteriocins in Health, Biotechnology, and Food Industry"

_antibiotics, 2024, doi:10.3390/antibiotics13070666_

Round 1

Reviewer 1 Report

Comments and Suggestions for Authors

The manuscript entitled “Bacteriocins, underestimated antimicrobials” is a review article that comparatively explains the bacteriocins potential as antimicrobial, their classification, types, production and application in the medical and food industry. The approach and comparative analysis of the present study is sound but still it needs some major revisions. The outcomes are interesting and the study will contribute to the knowledge in the respective area. However, the language revision should be painstakingly done. Some specific comments are as followed;

1. The title needs to be improved in writing format (e.g. multi-functional), if you targeting their antimicrobial potential against food and health by impacting their classification.   

2. Abstract needs to be revised. Clearly state the objectives of your review.  

3. Introduction section should be improved by adding some information on the antimicrobial potential of bacteriocins targeting biotechnological and pharmaceutical sector.

4. Add few lines on the classification basis, its problems and potentials in introduction section.

5. State the problem and your aims clearly in the introduction portion.

6. Why the post translational modifications based classification is important? Describe it clearly.

7. Lack of sufficient data about classification basis, their potentials and shortcomings.

8. Add some information on any potential drawbacks of already used bacteriocins? Their comparative analysis in food and health sector?  

9. Rationalize and define the future potentials of these products in antibiotics industry by targeting the potential harms.

10. Scientific names must be italic and abbreviated after their first use throughout the manuscript. A lot of such mistakes found in present draft.

11. The quality of all figures should be enhanced.  

12. Cross check for the spelling mistakes and grammatical errors. 

Comments on the Quality of English Language

Thorough proofreading of the article is recommended to eliminate any grammatical errors and typos and to ensure that the article is professional and readable.

Author Response

Dear reviewer.

Please find the reply to your suggestions on the attached PDF.

The authors thank you for your comments.

Reviewer 2 Report

Comments and Suggestions for Authors

1. This work lacks novelty as all figures were adapted from previously published works and few of those (for example, Figure 1) referred the original work as well. Moreover, the 'adaptation' is not enough to be considered as a representative figure or diagram. For example, only a smiley does not reflect 'Safety tests' (Figure 5)!

2. Resolution of all those figures are too low. I had to check the original figures present in the references to study the figures in this manuscript. And, authors should prepare the figure following the figures published in standard journals. For example, in Figure 3 authors should not write '3.1: Inhibition to cell wall synthesis' or '3.2: Pore formation' inside the figure. In addition, it did not match with the text. L121-138 shows both mechanisms (Disruption of cell wall synthesis and Pore formation under the sub-title of '3.1. LAB-Bacteriocins' but when we see the next sub-title '3.2. Colicins', it creates confusion if we try to understand Figure 3. Authors are strongly suggested to present their work clearly.

3. Authors are suggested to rewrite the manuscript with a formal approach. They should remove the 'Full stop' from the title, superscripted numbers from their names (as both authors are from the same institution) and that 'hyphen' joining their names. They numbered both the 'Perspectives' and the 'Conclusions' as 'Number 10'. L 288-291 are italicized with no obvious reason. Please edit the manuscript and present the data more carefully and clearly.    

Comments on the Quality of English Language

There are several sentences which are not very standard. Authors should take the help of a native English speaker to correct the manuscript. For example, check the legend of Figure 5. Or, L113: 3. 'Bacteriocins Mechanism of Action'.

Author Response

(The authors gave the same response as above.)

Reviewer 3 Report

Comments and Suggestions for Authors

There is a lot of similar data available related to the current topic.

Also, the information provided is not even sufficient to cover the field. 

All the figures or illustrations are adapted from others. 

Overall, there are major novelty and insufficiency issues with the current manuscript. 

Author Response

(The authors gave the same response as above.)

Round 2

Reviewer 2 Report

Comments and Suggestions for Authors

The manuscript looks improved. But it still lacks novelty as all eight figures are adaptations from previous articles. I expected at least one figure to be produced by the authors of this manuscript. Even a table could do the job. I strongly suggest to add a figure or table in this manuscript.

There are a few more suggestions:

1. I still can't see any 'comma' between the names of authors!

2. Figure 2: You put a '*' to point out that this classification was not recognized by all authors. Who are these authors? Authors of earlier articles? or, authors of this work? I suggest you to remove this information or put it in the figure legend (if you still want to keep it).

3. Serious grammatical mistakes are present in the 'Acknowledgements' and 'Author contributions'. Please correct those parts.

Comments on the Quality of English Language

Quality of English language is good. They improved the manuscript.

Author Response

Please find the author's reply on the attached file.

Reviewer 3 Report

Comments and Suggestions for Authors

The classification of bacteriocins represented in the manuscript is old and completely wrong.

Here is the most updated and globally accepted classification.

https://www.ncbi.nlm.nih.gov/pmc/articles/PMC3954855/ 

Also, the earlier issue mentioned by the reviewer is persisting. 

Author Response

(The authors gave the same response as above.)

Round 3

Reviewer 2 Report

Comments and Suggestions for Authors

Please check the Acknowledgement section. That sentence is still incorrect.

Comments on the Quality of English Language

Quality of English is alright.

Reviewer 3 Report

Comments and Suggestions for Authors

None.